# Surface Treatment Effects on the Mechanical Properties of Silica Carbon Black Reinforced Natural Rubber/Butadiene Rubber Composites

**DOI:** 10.3390/polym11111763

**Published:** 2019-10-27

**Authors:** Miaomiao Qian, Weimin Huang, Jinfeng Wang, Xiaofeng Wang, Weiping Liu, Yanchao Zhu

**Affiliations:** 1College of Chemistry, Jilin University, Changchun 130012, China; qianmm19@mails.jlu.edu.cn (M.Q.);; 2Deakin University, Institute for Frontier Materials, Geelong, Melbourne, VIC 3216, Australia; 3School of Chemistry, University of East Anglia, Norwich Research Park, Norwich NR4 7TJ, UK; 4College of Instrumentation & Electrical Engineering, Jilin University, Changchun 130012, China

**Keywords:** surface treatment, reinforcement fillers, rubber, phenolic resin, interfacial interactions

## Abstract

For the first time, phenolic formaldehyde resin (PF)-treated silica carbon black (SiCB) were prepared with different treatment conditions and their effect as fillers on the mechanical properties of filler filled natural rubber/butadiene rubber (NR/BR) composites were investigated in detail. The PF coating layer on the SiCB derived from rusk husk not only promoted the dispersion of the fillers but also improved the interfacial interactions between fillers and the rubber matrix. As a result, both the cross-link density and mechanical properties of the obtained composites were effectively enhanced. The filler SiCB with 3 wt % PF surface treatment greatly improved the tensile strength of NR/BR composites and reached 7.1 MPa, which increased by 73.7% compared with that of SiCB-filled NR/BR composites. The improved interfacial interactions promoted higher energy dissipation, leading to simultaneously enhancing the glass transition temperature of the obtained composites. Due to the easy processing and low cost of filler as well as the effectively enhanced mechanical properties of composites, the PF-coating methodology has a great potential for practical applications in SiCB reinforced high-performance composites. A commercial filler, carbon black (N774), was also used in this study and evaluated under the same conditions for comparison.

## 1. Introduction

Rice husk ash (RHA), obtained after pyrolysis the rice husks, has many advantages, such as being low cost, renewable, and having an environmentally-friendly character. Therefore, exploration of the potential application as rubber reinforcement at a big scale has attracted great interest. Early studies had been carried out [1,2], which revealed the possibility of applying ground RHA from a certain pyrolysis condition as moderately reinforcing filler for styrene-butadiene rubber (SBR), ethylene-propylene-diene elastomer (EPDM), and natural rubber (NR). RHA as filler did not affect either the vulcanization characteristics or the aging behavior of the obtained rubber composites. However, it was found that RHA provided lower mechanical properties to the obtained rubber composites (tensile strength, modulus, hardness, abrasion resistance, and tear strength) compared with other reinforcing fillers such as silica and carbon black [3,4,5].

It is worth noting that the mass content of silica in RHA in the reported work is above 80% [6]. Since silica in RHA has a high polarity and a hydrophilic surface due to silanol groups on its surface, RHA is incompatible with non-polar rubbers such as NR and is prone to aggregate. This property results in weak rubber-filler interaction and a poor filler dispersion in rubber, which limits the application of RHA as a reinforcement filler [7]. Many researchers have worked on new strategies to solve the problem, including adding coupling agents [8,9,10]. The interfaces were modified by the coupling agent, which can interact with both the hydroxyl groups on the silica surface and the rubber polymer during the compounding and vulcanization process, improving the interfacial interactions between RHA and the rubber matrix. However, the studied coupling agents are expensive for industrial application, and the application of these coupling agents involves the generation of volatile organic compounds (VOCs), such as methanol and ethanol, which is usually inevitable during the compounding process [11].

In our previous work [12,13], we prepared SiCB with a carbon content of around 50% through a pyrolysis process under 600–700 °C from rice husk. We have studied the SiCB itself in detail, including its physical and chemical properties, structure and possibility as a filler for rubbers. The comparison with carbon black has also been discussed. The obtained samples can make full use of rice husks, meanwhile a part of the char coat on the surface of the obtained silica particles during pyrolysis. Therefore, the number of exposed hydroxyl groups on the SiCB surface greatly reduced, preventing the agglomeration of silica and improving the compatibility of the filler in the polymer matrix. The mechanical and viscoelastic properties of composites filled with SiCB were improved compared with composites filled with RHA. To achieve reinforced polymer composites using reinforced filler, it is well known that the mechanical properties of the composites strongly depend on the interfacial adhesion between filler and polymer matrix [14,15,16] because the interface plays a critical role in stress transfer between the filler and the surrounding polymer matrix. SiCB needs to be treated to improve interfacial strength between SiCB and the rubber composites. Chemical treatment of filler surface is one of the main ways to improve dispersibility and interfacial interactions by introducing newly activated components or increasing the surface activity [17,18]. 

PF is one of the most commonly used resin materials because of its several desirable properties such as outstanding mechanical performance, high stiffness and hardness, reasonable flexibility, and low cost [19,20]. Many researchers have used PF as a reinforcing agent for rubber in recent years [21,22]. For example, PF was incorporated into acrylonitrile-butadiene rubber (NBR) vulcanizates by in situ polymerization during the vulcanization process. The mechanical properties of NBR vulcanizates were remarkably enhanced at the optimum PF content, which was only 15 phr (parts per hundred of rubber) [22]. Additionally, PF has also been used as a surface modifier for fillers to improve the reinforcement, while the unique three-dimensional network structure and chemical reactivity of PF provided a theoretical basis for its application as the surface modifier of fillers. Qing and You-Ping [23] found that the resorcinol-formaldehyde in situ treatment can efficiently improve the interfacial strength between starch and SBR in starch/SBR composites prepared by the latex compounding method. Compared to SBR, the mechanical properties of starch/SBR composite were greatly improved with only 1.2 phr. PF also was employed as a sizing agent to modify short carbon fiber (SCF) surfaces by a simple coating process, which improved the SCF/polyethersulphone (PES) interface adhesion with enhanced mechanical properties of the injection molded SCF/PES composites [24]. 

In this study, we report a facile route to modify SiCB with PF through surface melt coating. Namely, SiCB was added into molten PF to coat their surfaces with PF layers. The obtained SiCB/PF was used as a filler to reinforce natural rubber/butadiene rubber (NR/BR) blends. In order to study the surface coating effect of SiCB on the interfacial bonding between rubber and filler, the mechanical properties, viscoelasticity, and dynamic mechanical properties of the obtained rubber were investigated and discussed in detail. This work implicated the great potential of the developed SiCB/PF to be used as an efficient filler for NR/BR and also inspired the further design of the interface between filler and rubber.

## 2. Materials and Methods 

### 2.1. Materials 

Natural rubber (RSS1) consisting of cis-1,4-polyisoprene with a Mooney viscosity of ML (1 + 4) at 100 °C = 79.9, and butadiene rubber (BR9000) consisting of cis-1,4-polybutadiene with a Mooney viscosity of ML (1 + 4) at 100 °C = 50, were supplied by Shanghai Dukang Co., Ltd., (Shanghai, China). SiCB was produced by Jilin Kaiyu Biomass Development and Utilization Co., Ltd. (Changchun, China). Carbon black (N774) was obtained from Jiangxi Black Cat Carbon Black Co., Ltd. (Jingdezhen, China). Phenolic resin (2123) powder was supplied by Henan Xinxiang Co., Ltd. (Xinxiang, China). Stearic acid, zinc oxide (ZnO), and sulfur (S) in analytical grades were obtained from Sinopharm Chemical Reagent Co., Ltd. (Shanghai, China). All the other chemicals including *N*-1,3-dimethylbutyl-*N’*-phenyl-P-phenylenediamine (antioxidant 4020), poly(1,2-dihydro-2,2,4-trimethyl-quinoline) (antioxidant RD), wax, N-tertbutylbenzothiazole-2-sulphenamide (accelerator NS), and aromatic hydrocarbon oil (DAE) in chemical grade were supplied by Beijing Chemical Works. (Beijing, China) and used without further purification.

### 2.2. Preparation of PF-Treated SiCB 

SiCB was firstly added into a two-roller mixer (RM-200C, Harbin Hapro Electric Technology Co., Ltd., Harbin, China) with a speed of 40 rpm at different temperatures and held for a certain number of minutes. Then the determined amount of PF together with SiCB was gradually added into the mixer and mixed for 10 min.

### 2.3. Optimization of PF-Treated SiCB Preparation 

An orthogonal L_9_(4)^3^ test design was used for optimizing the PF-treated SiCB preparation conditions. Four variables at three different levels were investigated in the orthogonal design (Table 1). The stress level at 300% strain of the obtained NR/BR composites was used to determine the optimal preparation conditions of the fillers. The compounding formulations of the fillers are provided in Table 2.

### 2.4. Preparation of Filler-Rubber Composites

The rubber composites were prepared using a laboratory-sized internal mixer (KY-3220C-1L, Dongguan Kaiyan Machinery Equipment Factory, Dongguan, China) according to the formulations listed in Table 3. First, filler, antioxidants, wax, zinc oxide, and stearic acid were mixed with NR and BR in an internal mixer at 120 °C for 8 min. The composites were then taken out, cooled down to room temperature, and mixed with the accelerators and sulfur for 5 min at 100 °C. Finally, the obtained composites were cured into rectangular sheets with dimensions of 150 mm × 150 mm × 2 mm using compression molding at 150 °C for 10 min under 20 MPa. All the specimens for mechanical and physical tests were conditioned for 24 h before cutting. 

### 2.5. Characterization

#### 2.5.1. Microstructure

The surface morphology of SiCB was observed by a field emission scanning electron microscope (SU8020, HITACHI, Tokyo, Japan) at 20 kV. The samples were sputtered with gold to avoid electrical charging during testing. Fourier transform infrared (FTIR) spectra were carried out by 8400S (Shimazu, Kyoto, Japan) using the KBr method. The samples were ground with dry KBr in a mortar and then pressed to form thin disks. FTIR spectra were collected in the range of 450–4000 cm^−1^.

#### 2.5.2. Mechanical Properties

Mechanical properties were determined on a Universal Testing Machine (CMT-20, Jinan, China). Tensile testing was performed according to ISO 37: 2005 on two-type dumbbell specimens. Tear property was determined on right angle-type specimens according to ISO 34-1: 2004. Tensile and tear tests were carried out under a stretching speed of 500 mm/min at room temperature. The reported values are the average of at least five samples. 

#### 2.5.3. Mullins Effect

Cyclic loading, unloading, and reloading uniaxial tension tests were performed to obtain the viscoelastic non-linear behavior of the composites. A cyclic tensile test was performed on three samples at a speed of 60 mm/min. The samples were elongated to 100% strain with a peak displacement of 40 mm.

#### 2.5.4. Dynamic Mechanical Analysis (DMA)

The dynamic mechanical thermal properties of composites were analyzed by dynamic mechanical thermal analyzer (Q800, TA, New Castle, Delaware, America) at 10 Hz in the tension mode with a strain amplitude of 0.2%. The test temperature ranged from −80 °C to 40 °C with a heating rate of 5 °C/min. Composites were cut into rectangular bars with a dimension of 48 mm × 6 mm × 2.5 mm.

## 3. Results and Discussion

### 3.1. Optimal Filler S/P-X Preparation Process

A number of variables were investigated using orthogonal experimental design, including packing amount in the mixing chamber, mixing temperature, mixing time, and PF content. Three different levels of the four variables were chosen within a certain range to investigate their effect on the mechanical properties of the obtained composites. 

As shown in Appendix A, the highest stress at 300% strain of composites was 3.1 MPa, which was achieved at test 9. The m and R values were calculated and listed in Appendix A based on orthogonal experiment and range analysis. R decreases in the order of mixing temperature > PF concentration > mixing time > packing amount. This suggests that the influence on stress at 300% strain decreases in the order of mixing temperature > PF concentration > mixing time > packing amount. In other words, mixing temperature plays the most important role in achieving high stress at 300% strain of S/P-X-filled NR/BR composites. The combination of the levels with the highest m value for each variable gives rise to the optimum conditions. It can be seen from Appendix A (highlighted in bold) that the optimum conditions for stress at 300% strain are the combination of packing amount of 180 g, treatment temperature of 100 °C, the treatment time of 10 min and PF concentration of 3%. Therefore, a new sample (S/P-100) was prepared according to the obtained optimized conditions and used for further characterization. 

The optimum temperature was further confirmed by single-factor experiments. The fillers (preparation conditions are listed in Appendix A) were filled into the NR/BR composites and the mechanical properties of the composites were tested. As shown in Figure 1a, the tensile strength of the obtained composites increased as the temperature increased and reached the highest of 7.1 MPa at 100 °C. Meanwhile, stress at 300% strain also reached the maximum at 100 °C and then decreased with further increase of the temperature. Figure 1b represented the effect of phenolic resin treated SiCB at different temperatures on tear strength and elongation at the break of NR/BR composites. Both the tear strength and elongation at break achieved the highest value when the filler was prepared at 100 °C. These results could be attributed that PF melted and dispersed evenly at 100 °C, which acted as the bridge effect between SiCB and NR/BR and improved the interfacial interactions between SiCB and rubber polymers. This led to improved mechanical properties of the obtained rubber composites.

### 3.2. Cross-Link Density

The mechanical properties of rubber closely relate to the concentration and distribution of the effective network chain. The cross-link density represents the number of cross-link points per unit volume, which reflects the number of effective network chains per unit volume [25,26]. Therefore, within a certain range, the higher the cross-link density is, the better the mechanical properties will be that are achieved. In this work, the cross-link density was determined from the stress-strain curves according to Equations (1) and (2) via the Mooney–Rivlin approach [27,28,29].
(1)$$\sigma =2\left(c_{1}+\frac{c_{2}}{\lambda }\right)\left(\lambda -\frac{1}{\lambda^{2}}\right)\;\mathrm{with}\;\lambda =1+X\varepsilon$$
(2)$$\upsilon =\frac{2c_{1}}{k_{B}T}$$
where σ is the true stress measured in the strained state, C_1_ and C_2_ are characteristic Mooney–Rivlin parameters of cross-linked rubber, representing the effects of chemical cross-links and entanglements, respectively, λ is the extension ratio, X is the strain amplification factor defined as σE_0_/ε (X = 1 for gum rubber), ε is the engineering strain (the ratio between increased length and initial length during the testing ($\varepsilon =\Delta L/L_{0}$)), υ is the cross-link density, k_B_ is the Boltzmann constant (1.38 × 10^−23^ m^2^ kg s^-2^ K^−1^), and *T* is the temperature (K).

The υ values of NR/BR composite filled with S/P-X are shown in Figure 2 and summarized in Table 4. It can be seen that the υ values gradually increased with increasing the temperature and reached the largest value of 2.24 × 10^–4^ mol/cm^3^ for the sample with filler S/P-100. The υ started to decrease with further increasing the temperature. The results indicated that the cross-link density was related to the state of PF coating. At 100 °C, SiCB was coated uniformly with a thin layer of PF, which provided a stronger interaction through chemical bonds with NR molecular chains [30], leading to improved cross-link density. Therefore, PF can strengthen the interfacial bonding and link more closely between rubber and filler.

### 3.3. Characterization of the Fillers

The coating of PF on SiCB was further verified by FTIR spectra as shown in Figure 3a. For the pristine SiCB, the strong broad band at approximately 1100 cm^−1^ is assigned to Si–O vibration [31]. The interaction between PF and SiCB is supported by the appearance of new peaks. The small peaks around 2800–3100 cm^−1^ are attributed to the –CH stretching, methylene of –CH_2_– and dimethylene ether bridges of –CH_2_OCH_2_–. The narrow peaks at 1515 cm^−1^ and 1465 cm^−1^ are assigned to the C=C aromatic ring, methylene bridges, and the C–OH asymmetric stretching vibration. The absorption peak at 728 cm^−1^ can be assigned to ring deformation [32,33]. 

To investigate the temperature effect on PF coating on the surface of SiCB, an SEM was used to observe the surface morphology of the obtained filler S/P-80, S/P-100 and S/P-120 as shown in Figure 3b–d. A discontinuous white layer was observed on the surface of S/P-80 as seen in Figure 3b. This indicated that the phenolic resin was not completely fused, which resulted in the agglomeration of the SiCB particles. For S/P-100, a uniform and nearly fully coating of PF was observed on the surface of SiCB. When the temperature was further increased to be 120 °C, a part of the PF layer peeled off from the surface of the SiCB (Figure 3d). These results demonstrated that the optimal coating of PF on the SiCB surfaces was achieved at 100 °C. Therefore, the filler S/P-100 was prepared under the obtained optimized conditions and used for further characterization.

### 3.4. Mechanical Properties

The optimal filler S/P-100 effects on the mechanical properties of the obtained NR/BR composites were investigated. The mechanical properties of RHA-, SiCB-, and commercial carbon black (N774)-filled NR/BR composites were also carried out as comparison as shown in Figure 4. S/P-100 filled NR/BR showed the highest tensile strength compared to all the other obtained composites, which reached up to 7.1 MPa. This was compared with 6.5 MPa for semi-reinforcing carbon black (N774)-filled NR/BR, 4.1 MPa for SiCB-filled NR/BR and 1.9 MPa for RHA-filled NR/BR. The stress at 300% strain of S/P-100 filled NR/BR composites showed a 16.8% increase compared with SiCB-filled NR/BR composites.

The mechanical properties of these obtained composites were also summarized in Table 5. It can be seen that S/P-100 filled composites showed improved properties compared to N774-filled composites except for stress at 300% strain. Compared to SiCB-filled composited, S/P-100 filled composites significantly improved the tensile strength by two times. The improvement could be attributed to the strong interface between filler S/P-100 and matrix, which further proves that the effect of the PF coating. This result indicated that S/P-100 possesses great potential to be used as a semi-reinforcing filler for rubber.

The elongation at the break of a composite is strongly associated with the filler networks formed in the composite. Figure 4b illustrated the elongation at the break of NR/BR composites filled with different fillers. The existence of PF coating greatly improved the elongation at the break of the obtained composites, corresponding to an increase of 71.3% compared to SiCB-filled composites. This increase of the elongation at break is attributed to the improved interfacial interactions between NR/BR and S/P-100, which greatly decreased the aggregation of SiCB and PF could increase the slippage of SiCB in the polymer chains when an external force is applied [34].

To verify the effect of PF coating on the mechanical performance of the obtained composites, PF was added directly in the process of mixing to prepare SiCB/PF-NR/BR composites. As shown in Appendix A, SiCB/PF-NR/BR composites showed a lower tensile strength of 5.7 MPa, which was compared with 7.1 MPa for S/P-100 filled NR/BR composites. This result further confirms the surface melt coating achieved a good dispersion of PF on the surface of SiCB, leading to enhancement of SiCB and rubber composites interfacial strength and increased composite mechanical performance.

### 3.5. Mullins Effect

Compounding elastomeric materials with fillers provides not only high mechanical performance but also high inelastic behavior. Many of the inelastic models have been developed under specific loading conditions such as small strain, high strain rate, cyclic loading, or when subjected to softening behavior [35,36,37]. An important phenomenon is the Mullins effect, observed at large strains. It includes the cyclic stress softening and the ability to recover and return towards the virgin stress-strain path when the load rises over the previous maximum applied stress value. We suggest that there are several interactions involved in the stress softening resulting from the Mullins effect, which includes breakage or slippage of network chains connecting the filler particles [38,39,40], breakage of the filler aggregates [41], and macromolecular disentanglement [42]. To the best of our knowledge, no work has been conducted to investigate the Mullins effect of rubber composites filled by RHA. Figure 5 shows the stress-strain curves of vulcanized NR/BR systems with different fillers under cyclic uniaxial tension. All three composites showed stress softening during tensile loading and unloading cycles. The hysteresis, estimated as the area between the loading and the unloading stress-strain paths, decreased with increasing the cycles of tensile. The area is generally defined as the energy dissipation of material to overcome internal friction [43]. Energy dissipation is calculated based on the equation presented in previous work [13]. It can be seen that the hysteresis loop was large in the first cycle and became smaller in the second cycle, and then maintained after a few repeated cycles. This is consistent with the Mullins effect [34].

All three composites were subjected to cyclic uniaxial tension and the dissipated energy at different cycles was shown in Figure 6a. It can be seen that the energy dissipation of SiCB-filled composites was larger than that of N774-filled composites under 100% strain. It may be caused by the strong filler-filler networks resulting from the poor dispersion of SiCB [44]. However, the addition of 3 wt % of PF (S/P-100 filled composites) greatly increased the energy dissipation, which indicated a significant increase of the cross-link points and the interfacial interactions between fillers and NR/BR. Therefore, the surface coating of SiCB played an important role in viscoelasticity of the obtained rubber. The surplus ratio of the energy dissipation of all three composites after several reciprocations was calculated as shown in Figure 6b. For S/P-100 filled composites, the surplus ratio was the largest among the three composites. This could be caused by the improved slippage of SiCB particles in the rubber chains in the presence of PF, which reduced the filler-filler friction and thus increased the surplus ratio of the energy dissipation.

### 3.6. Dynamic Mechanical Analysis

DMA can provide information about the dynamic mechanical properties of the materials [45,46]. In this study, DMA of the SiCB, S/P-100 and N774 filled NR/BR composites was carried out to evaluate the interfacial adhesion in polymer matrix composites. The temperature dependence of the storage modulus, *logE’*, of NR/BR composites filled with three different fillers and 1^st^ derivative of storage modulus curves were displayed in Figure 7. The glass transition temperature (*T*_g_) was measured at the peaks of the first derivative of storage modulus curves. We found the *T*_g_ of S/P-100 filled composites showed an increase of 3 °C compared to that of SiCB, and was higher than that of N774-filled composites. This was attributed to the effect of the surface coating of SiCB on the interfacial bonding between rubber and filler, which reduced the filler-filler networks in the composites, and the number of the macromolecular chains entrapped in the filler clusters decreased. Therefore, more rubber chains interacted with the fillers, leading to increased *T*_g_.

## 4. Conclusions

In this work, PF was successfully coated on SiCB and optimized through an orthogonal experimental design. The PF-treated SiCB was then used as a filler to prepared NR/BR composites through a two-roller mixer followed by compression molding using hot press. The PF coating on the surface of SiCB not only increased the surface activity of SiCB, but also improved the interfacial interactions between SiCB and NR/BR composites. Compared to SiCB-filled NR/BR, the optimized S/P-100 filled NR/BR greatly improved the tensile strength, stress at 300% strain and the tear strength for 73.7%, 16.8%, and 21.1%, respectively. The Mullins effect of S/P-100 filled NR/BR also showed higher energy dissipation before failure compared to SiCB filled NR/BR. Moreover, the *T*_g_ of S/P-100 filled composites showed an increase of 3 °C over that of SiCB. This work shows the potential of using PF as a promising surface modifying agent for SiCB to improve the interfacial strength between the filler and rubber to effectively enhance the mechanical properties of NR/BR composites.

## Figures and Tables

**Figure 1 polymers-11-01763-f001:**
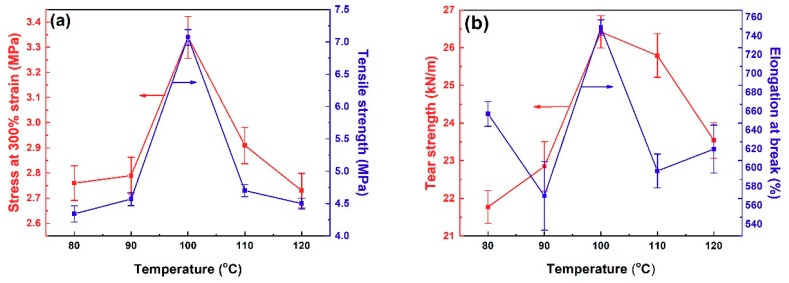
Filler treatment temperature effect on the mechanical properties of filler-filled NR/BR composites: (**a**) tensile strength and stress at 300% strain; (**b**) tear strength and elongation at break.

**Figure 2 polymers-11-01763-f002:**
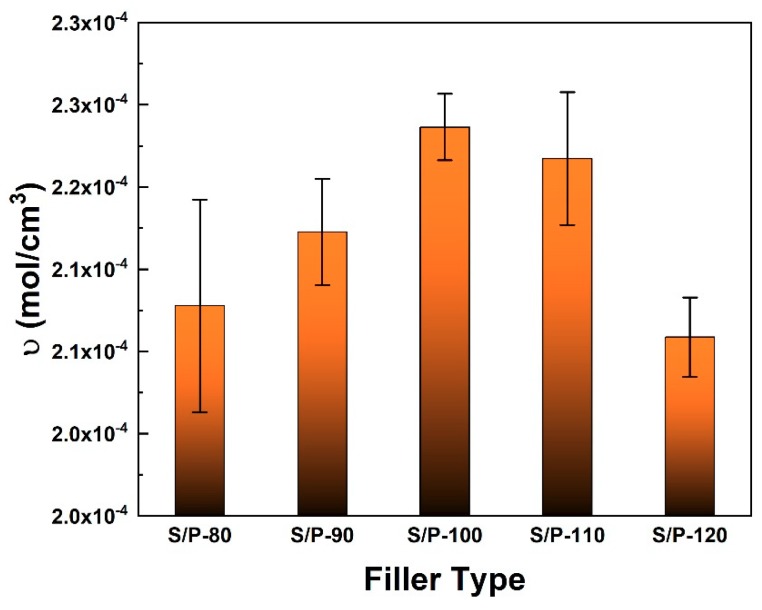
The υ values of NR/BR composites filled by different S/P-X fillers.

**Figure 3 polymers-11-01763-f003:**
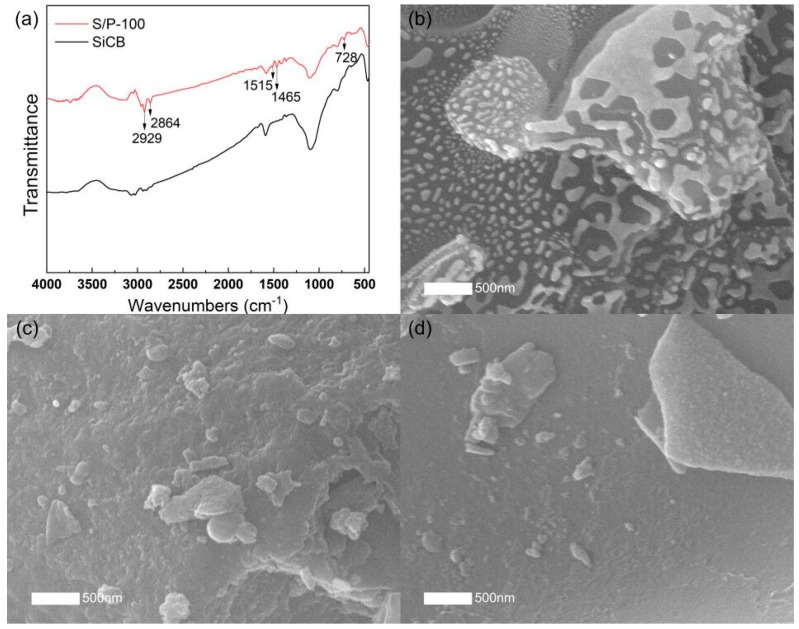
FTIR spectra for SiCB and S/P-100 (**a**). SEM images of (**b**) S/P-80, (**c**) S/P-100 and (**d**) S/P-120.

**Figure 4 polymers-11-01763-f004:**
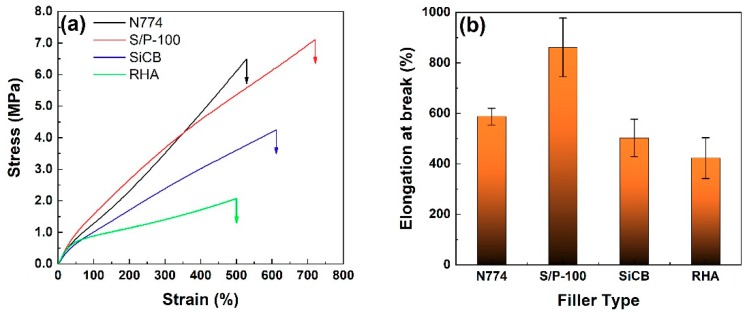
(**a**) Representative stress-strain curves of NR/BR composites filled with different fillers. (**b**) Elongation at break of NR/BR composites filled with different fillers.

**Figure 5 polymers-11-01763-f005:**
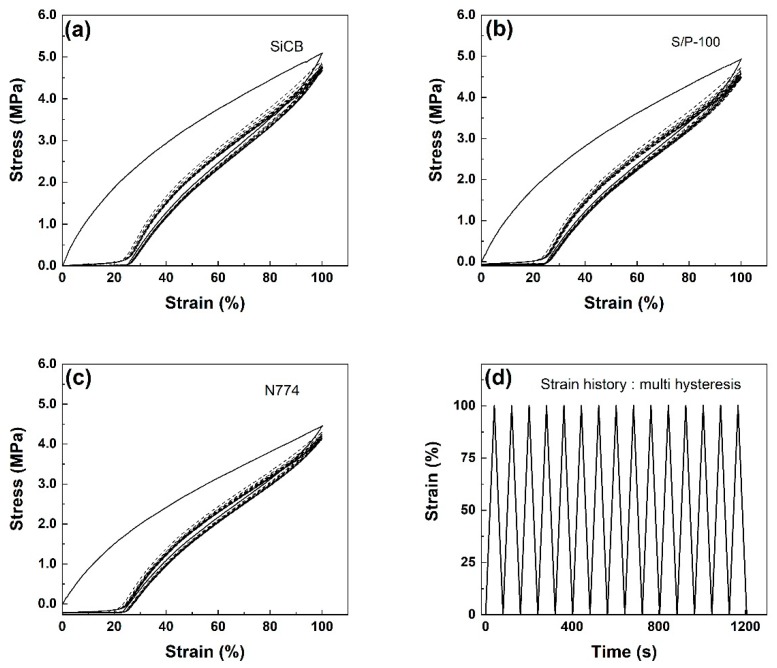
Stress-strain response of NR/BR filled with SiCB (**a**), S/P-100 (**b**) and N774 (**c**) submitted to cyclic uniaxial tension. Cyclic strain history with constant strain rate and strain amplitudes organized in step up and step down. The samples were loaded with fifteen fully relaxing cycles (**d**).

**Figure 6 polymers-11-01763-f006:**
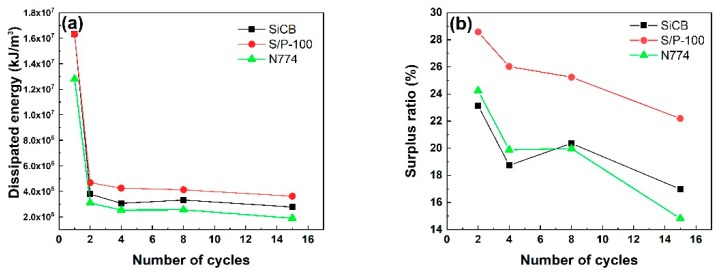
(**a**) Dissipated energy at different cycles for all the three composites subjected to cyclic uniaxial tension. (**b**) Surplus ratio of dissipated energy at different cycles for all three composites subjected to cyclic uniaxial tension. The surplus ratio was calculated as the value of residual dissipated energy divided by the dissipated energy in the first cycle.

**Figure 7 polymers-11-01763-f007:**
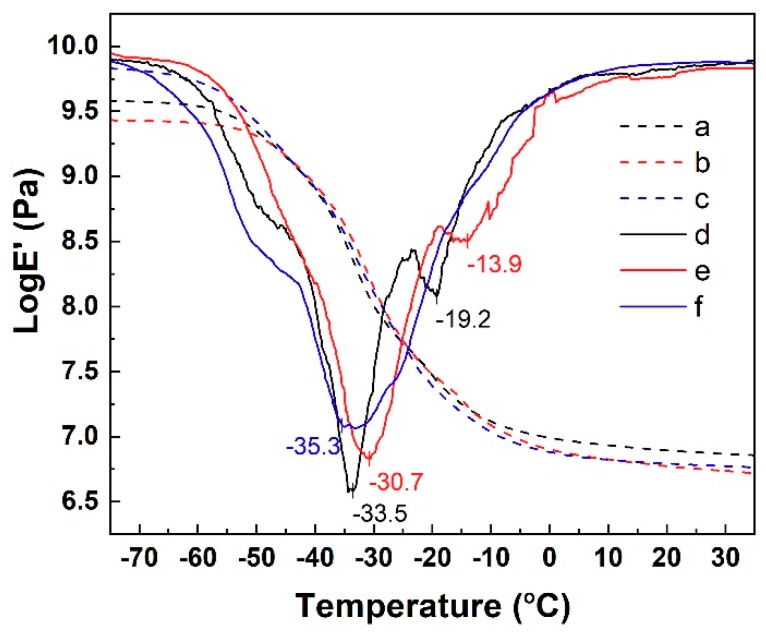
Temperature dependence of the *logE’* value of (**a**) SiCB, (**b**) S/P-100 and (**c**) N774-filled NR/BR composites. Temperature dependence of 1^st^ derivative of storage modulus curves of (**d**) SiCB, (**e**) S/P-100 and (**f**) N774-filled NR/BR composites. The glass transition temperature (*T*_g_) was measured at the peak of 1^st^ derivative of storage modulus curves.

**Table 1 polymers-11-01763-t001:** Treatment conditions.

Factor	Packing Amount (g)	Temperature (°C)	Time (min)	PF Concentration (wt %)
Level 1	160	90	5	3
Level 2	170	95	10	5
Level 3	180	100	15	8

**Table 2 polymers-11-01763-t002:** Compounding formulation of the fillers.

Abbreviation	Composition and Treating Condition		
	Raw Material	Phenolic Resin or Not	Treatment Condition
SiCB	Rice husk biochar	No phenolic resin	Untreated
SiCB/PF	Rice husk biochar + phenolic resin	Phenolic resin	Untreated
S/P-X ^a^	Rice husk biochar + phenolic resin	Phenolic resin	80–120 °C
RHA	Rice husk ash	No phenolic resin	Untreated
N774	Commercial carbon black	No phenolic resin	Untreated

^a^ X represents the treatment temperature.

**Table 3 polymers-11-01763-t003:** Rubber formulations.

Ingredients	Amounts (phr)
NR/BR	45/55
Filler	50
ZnO	3
Stearic acid	1.5
Wax	1.5
DAE	10
Antioxidant RD	1.5
Antioxidant 4020	4
Accelerator NS	0.75
Sulfur	1.8

**Table 4 polymers-11-01763-t004:** The υ values of NR/BR composites filled with S/P-X.

Fillers	C_1_ (10^−2^ MPa)	C_2_ (10^−2^ MPa)	υ (10^−5^ mol/cm^3^)
S/P-80	26.3 ± 0.8	16.7 ± 0.7	21.3 ± 0.6
S/P-90	26.9 ± 0.4	21.2 ± 0.4	21.7 ± 0.3
S/P-100	27.7 ± 0.3	15.3 ± 0.2	22.4 ± 0.2
S/P-110	27.4 ± 0.5	18.7 ± 0.6	22.2 ± 0.4
S/P-120	26.1 ± 0.3	19.1 ± 0.4	21.1 ± 0.2

**Table 5 polymers-11-01763-t005:** Mechanical properties of RHA-, SiCB-, S/P-100-, and carbon black (N774)-filled NR/BR composites.

Fillers	Tensile Strength (MPa)	Elongation at Break (%)	Stress at 300% Strain (MPa)	Tear Strength (kN/m)
RHA	1.9 ± 0.04	422.7 ± 80.7	1.5 ± 0.1	15.0 ± 0.4
SiCB	4.1 ± 0.1	502.9 ± 74.3	2.9 ± 0.3	21.8 ± 0.4
S/P-100	7.1 ± 0.1	861.5 ± 116.2	3.3 ± 0.4	26.4 ± 0.5
N774	6.5 ± 0.1	587.1 ± 33.1	3.7 ± 0.4	27.4 ± 0.6

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
