# Peer review of "Surface Treatment Effects on the Mechanical Properties of Silica Carbon Black Reinforced Natural Rubber/Butadiene Rubber Composites"

_polymers, 2019, doi:10.3390/polym11111763_

Round 1

Reviewer 1 Report

I would like to report on the manuscript of Qian et al. entitled “Effects of SiCB surface modification with phenolic resin on mechanical properties of natural rubber/butadiene rubber composites”. The authors report an interesting polymer application in the field of rubber reinforcement. The authors describe a process to yield a phenolic resin-modified silica carbon black (SiCB) initially obtained from rice husk ash. The surface modification of the particles drives an efficient reinforcement of a rubber sample (natural rubber/butadiene rubber) as demonstrated, by an increase in tensile strength for the composite material among other mechanical data. The introduction describes the context and the subject clearly enough, the experiments of the paper are well conducted and the experimental results look promising.

My main point of criticism is that the paper deals with a highly applied chemical science’s subject as exemplified by the fact that most starting products have not a well-defined structure and have not been characterized extensively as regards molecular weights, structure and composition : natural rubber, butadiene rubber, phenolic resin. This criticism does minimize the quality of the manuscript, but makes me believe that it is not suitable with the main guidelines of “Polymers”. In my opinion, and based on my understanding of the editorial line of “Polymers”, this paper would be more suitable for an applied polymer journal such as J. Appl. Polym. Sci. (highly cited in many references) or J. Compo Part A. I personally consider “Polymers” has a general polymer journal, which requires a minimum of information on the starting materials, but also the chemical reactions taking place. For example, the surface characterization carried out using FTIR analysis and SEM in the present paper turns out to be too limited. Despite the interest and quality of the manuscript, I recommend in conclusion a new submission in a more specialized journal. Below are recommendations to improve the manuscript, mostly in the introduction section.

ARTICLE TITLE:

The term SiCB (Silica carbon black) should be defined in the title to make the paper accessible to a broader audience

INTRODUCTION:

L34: The authors should explain the composition and structure of rice husk ash (RHA), in particular comparison with more conventional rubber fillers such as silica or carbon black should be an added value. L35: How does the thermal treatment of rice husk affect the final content of carbon and silica ? L35 : “environmental amity” should be replaced by “environmentally friendly”. L35 : “in big scale” should be replaced by “at large scale”. L 39 “RHA as the filler” replaced by “RHA as filler”. 11 is not clear. “Wang, X.F.; Zhou, Y.; Zhu, Y.C. Preparation method of SiCB/polyolefin compounds, 201610600982.7, 2016-396 07-28”. The differences with previous works (ref. 11 and above ref. 12) should be clearly exposed and clarified in the introduction section. L 69 : authors should clearly state the type of phenolic resin used in the different cited publications. The phenol resin structure can obviously affect the final properties of the composite material. L 74. « optimum PF content of only 15 phr (per hundred rubber). » Such expression of concentration is not clear. L 84 : “surface melt coating” treatment should be explained. L 86: authors should justify the use of NR/BR as reference rubber in this system. A mono-component rubber might have been more appropriate for a feasibility study.

Reviewer 2 Report

This paper studied the effect of phenol/form aldehyde resin on the modification of SiCB filler from rice husk. Authors emphasized that the modification condition is best at 100℃, 10 min of mixing time, and 3% concentration of PF resin.

However, there are a lot of question on the experiments and analysis of test data as follows:

The shape (or morphology), size, and structure should be reported for the fillers of RHA, SiCB, S/P-100, and N774. A detailed graphical configulation for S/P-100 filler should be suggested. Line 84-92: The purpose of this paper should be written, not for the experimental results. Figure 2: The values of Mc are too high. Usually these values are 4,000-10,000 gr/mole. Also the plotting of Mooney-Rivlin equation should be suggested. Figure 3: Is the shape of S/P-100 plate-like? Table 4: The data of elongation at break and E300 are not consistent with Fig 4(a). Section 3.5 & 3.6: The explanation is unclear. The End -

Reviewer 3 Report

In my opinion this paper is interesting. However, before the final decision manuscript should be carefully revised by authors. I suggest major revision including improvement quality for most of presented figures.

My detailed comments in the attachment.

Round 2

Reviewer 1 Report

The authors have made significant efforts to improve the manuscript and answer all of the referee's questions, which is a positive point.

However, my initial position on the manuscript has not changed, because my decision is based on the general content of the manuscript and not only on technical points (originality, quality of presentation, scientific soundness, etc.). I believe that the document would be more suitable to another more applied journal for different reasons mentioned again below:

The paper deals with a highly applied chemical science’s subject. Most starting products have not a well-defined structure and have not been characterized extensively as regards molecular weights, structure and composition : natural rubber, butadiene rubber, phenolic resin. Most cited papers come from highly applied polymer journal such as J. Appl. Polym. Sci. or J. Compo Part A. Lack of information on chemical reactions.

Despite the interest and quality of the manuscript, I recommend a new submission in a more specialized journal, but let the final word to the editor because he is best positioned to decide if the document complies with the editorial line of the journal.

Reviewer 2 Report

This manuscript is very difficult to read because readers cannot understand the state of fillers, like filler size, surface area, shape, and morphology, etc., for SiCB, S/P-100, and N774 fillers used in this manuscript. Authors should present the above data. Also authors should suggest graphical diagrams for the fillers.

For the section 3.5 Mullin’s effect, authors emphasized that the stress softening is due to the breakdown of filler network. However, the stress softening is mainly due to the chain breakage and slippage of molecules. (Ref: Science and Technology of Rubber, 2nd ed., by Academic Press, p407-411). If the authors maintain the breakdown of filler network, please suggest reference papers.

The End -

Reviewer 3 Report

In my opinion this paper can be considered for publication after minor review.

Detailed list of comments in the attachment.

Round 3

Reviewer 2 Report

Authors did good job for the improvement of this manuscript according to my comments.

I agree to publish this manuscript in the journal of Polymers.

- The End -